# Conversion of Dimeric Diaryl Ethers over SiO₂- and HZSM5-Supported Pd and Ru Catalysts: A Focus on the Role of the Metal and Acidity

Raphaela Azevedo Rafael [1], Fabio Bellot Noronha [1,2], Eric Marceau [1] and Robert Wojcieszak [1,*]

[1] University of Lille, CNRS, Centrale Lille, University of Artois, UMR 8181—UCCS—Unité de Catalyse et Chimie du Solide, F-59000 Lille, France; raphaela.azevedorafael.etu@univ-lille.fr (R.A.R.)

[2] Catalysis Division, National Institute of Technology, Av. Venezuela 82, Rio de Janeiro 20081-312, Brazil

[*] Correspondence: robert.wojcieszak@univ-lille.fr; Tel.: +33-(0)320676008

**Abstract:** The effect of metal and support acidity on the hydroconversion of dimeric aryl ethers, used as model molecules for lignin, is still under debate, both in terms of hydrogenolysis (cleavage of the ether bond) and formation of by-products (coupling of aromatic monomers to dimers by alkylation reaction). Their role is investigated here in the conversion of three typical molecules representative of the α-O-4, β-O-4, and 4-O-5 ether linkages of lignin, respectively, benzyl phenyl ether (BPE), phenethoxybenzene (PEB), and diphenyl ether (DPE), at 503 K, under 18 bar of H₂ in decalin. Ru- and Pd-based catalysts were synthesized on non-acidic SiO₂ and on acidic HZSM5. Under these reaction conditions, the conversion of the ethers over the bare supports was observed in the presence of acidic sites; the effect decreased as the ether bond strength increased. The results also suggest that the product distribution is directly affected both by the support acidity and by the oxophilicity of Ru. Alkylated products from isomerization reactions, which are reported to be formed only over acidic sites, were also produced on the surface of the Ru nanoparticles.

**Keywords:** hydrogenolysis; alkylation; benzyl phenyl ether; phenethoxybenzene; diphenyl ether



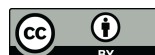

## 1. Introduction

Lignin is the most stable fraction of biomass, and it is considered to be an important resource for the production of high-added-value chemicals and fuels [1,2]. However, due to its complex structure, lignin is currently used as a low-grade fuel to provide heat, with low efficiency and, consequently, with an increase in greenhouse gas emissions [3–5]. Therefore, numerous efforts are being made to promote lignin valorization through technological advancement for the realization of sustainable resource management [4].

Lignin is an amorphous polymer composed of phenylpropane units randomly connected via C-C and C-O linkages. The three basic monolignols that constitute the lignin structure are p-coumaryl, coniferyl, and sinapyl alcohol. The concentration of these phenylpropane units varies according to the type of biomass, type of tissue, as well as with the growing conditions of the plant [6,7].

Several approaches have been used for the conversion of biomass into transportation fuels: among them, gasification to syngas, hydrolysis to monomeric units, and pyrolysis to produce bio-oil [8,9]. However, the presence of oxygen in the resulting compounds decreases the calorific capacity of bio-oil and makes it thermally and chemically unstable. To be used directly as a fuel or mixed with crude oil, bio-oil must be upgraded by catalytic hydroconversion. The hydrodeoxygenation reaction (HDO) is considered the most effective process. Oxygen present in bio-oil can be effectively removed in the presence of a heterogeneous catalyst and a hydrogen source. In general, high temperature and hydrogen pressure are required to achieve complete deoxygenation.

As bio-oil is a complex mixture, dimeric aryl ethers such as benzyl phenyl ether (BPE), phenethoxybenzene (PEB), and diphenyl ether (DPE), representative of the main ether linkages present in lignin (α-O-4, β-O-4, and 4-O-5 linkages, respectively), have been reported in the literature as model compounds for the conversion of lignin and bio-oil [10–18]. Regarding the bond strength, the 4-O-5 ether linkage of DPE (bond dissociation energy (BDE) = 314 kJ·mol$^{-1}$) is much stronger than the aliphatic ether bond β-O-4 (289 kJ·mol$^{-1}$) and α-O-4 (218 kJ·mol$^{-1}$) [19]. Different catalysts and reaction conditions have been used to transform these model molecules into aromatics or cycloalkanes. Metals such as Ru, Pd, and Ni supported on SiO$_2$, carbon, and zeolites have been extensively studied for conversion [11,12,14,19–28]. However, a systematic study that compares metals over supports with different properties under the same reaction conditions is still missing, and it is difficult to discriminate what effect the metal and the support properties have on the transformation of these ethers.

For instance, in the conversion of BPE over Ni/SiO$_2$ [12] (523 K and 40 bar of H$_2$), BPE was converted into phenol and toluene via hydrogenolysis catalyzed by the metallic sites. Depending on the reaction conditions, phenol can be hydrogenated to cyclohexanone/cyclohexanol, and in the presence of acidic sites, cyclohexanol can be dehydrated into cyclohexene, which can be transformed into cyclohexane via its hydrogenation [20,23,29]. However, the formation of alkylated products originating from the isomerization of the ether was also reported and assigned to the presence of acidic sites [12,30].

For the conversion of PEB over Pd, Ru, and Ni supported on activated carbon at 423 K and under 20 bar of H$_2$, Ni and Pd exhibited the highest ability to break the C-O ether bond of PEB into ethylbenzene and phenol by hydrogenolysis; in contrast, Ru promoted mainly the hydrogenation of PEB aromatic rings [25]. However, at 513 K and under 8 bar of H$_2$, reported results were different: Ru/C and Pd/C exhibited higher activity for the C-O cleavage, but phenol hydrogenation was favored on Pd/C. In addition, the hydrogenation of PEB aromatic rings was not observed [11]. In the case of this ether, the alkylation of PEB was only reported in a few works [14,31].

Regarding the conversion of DPE, Ru, Pd, and Pt supported over activated carbon were also used by Wu et al. [32] to convert DPE at 393 K under 5 bar of H$_2$. The authors observed that Ru exhibited the highest conversion and activity in the C-O bond cleavage, while Pd and Pt showed the highest activity in the hydrogenation of DPE aromatic rings. Zhao et al. [20] used acidic Ni/HZSM5 and Pd/HZSM5 for the conversion of DPE at 493 K under 20 bar of H$_2$. While Pd/HZSM5 converted 70% of DPE into cyclohexane, dicyclohexyl ether, and cyclohexyl phenyl ether, Ni/HZSM5 did not exhibit any conversion. However, this result diverges from those from Zhao and Lercher [22], who also used Ni/HZSM5 for the conversion of DPE (523 K, 50 bar H$_2$) but obtained 64% of DPE conversion with high selectivity to cyclohexane.

It can be clearly seen from above that even if the catalysts are similar, the reaction conditions are always different. From the existing literature, it is still difficult to delineate precisely the role played by the type of metal and the role played by the support properties, such as its acidity, on the parallel and successive steps of the hydroconversion of dimeric aryl ethers under hydrogen pressure.

Therefore, the present work aims at investigating the role of the nature of the metal, as well as the effect of the support acidity, in the cleavage of model molecules representative of the α-O-4, β-O-4, and 4-O-5 ether linkages. Pd, and a metal known for its more oxophilic character toward oxygen-containing substrates, Ru, were selected as the metallic phase, while support without acidic sites, SiO$_2$, and acidic support, HZSM5, were investigated on the conversion of BPE, PEB, and DPE.

## 2. Results

### 2.1. Catalysts Characterization

Table 1 reports the results of chemical composition, specific surface area, and pore volume of Pd- and Ru-based catalysts supported over SiO$_2$ and HZSM5. The Pd contents

were close to the expected values (2 wt.%); however, the Ru contents varied from 0.6 to 0.7 wt.% and were lower than the expected value (1 wt.%). The specific surface area of the supported catalysts did not show significant changes after the impregnation and calcination steps. The $N_2$ adsorption–desorption isotherms of the bare supports, as well as of the Pd- and Ru-based catalysts, are presented in Figure 1. $SiO_2$ and its respective supported catalysts displayed a type IV isotherm characteristic of mesoporous materials with a type H1 hysteresis loop that originates from the non-rigid aggregates of plate-like particles. For HZSM5, Pd/HZSM5, and Ru/HZSM5, a type I isotherm, typical of microporous materials, was observed [33].

**Table 1.** Pd and Ru content, specific surface area, pore volume, particle size and dispersion of palladium and ruthenium-supported catalysts.

| Catalysts | Metal Loading (wt.%) | BET ($m^2 g^{-1}$) | Pore Volume ($cm^3 g^{-1}$) | Dp (nm) [c] | D (%) [e] |
|---|---|---|---|---|---|
| $SiO_2$ | - | 189 | 0.76 | - | - |
| Pd/$SiO_2$ | 1.8 [a] | 182 | 0.90 | 7.2 (17) [d] | 15 |
| Ru/$SiO_2$ | 0.6 [b] | 190 | 0.72 | 8.3 (10) [d] | 16 |
| HZSM5 | - | 485 | 0.33 | - | - |
| Pd/HZSM5 | 1.8 [a] | 393 | 0.39 | 24.6 (22) [d] | 5 |
| Ru/HZSM5 | 0.7 [b] | 518 | 0.34 | 12.0 (15) [d] | 11 |

[a] Measured by ICP. [b] Measured by XRF. [c] Measured by TEM. [d] Crystallite diameter estimated from XRD data by Scherrer equation. [e] Calculated based on TEM analysis.

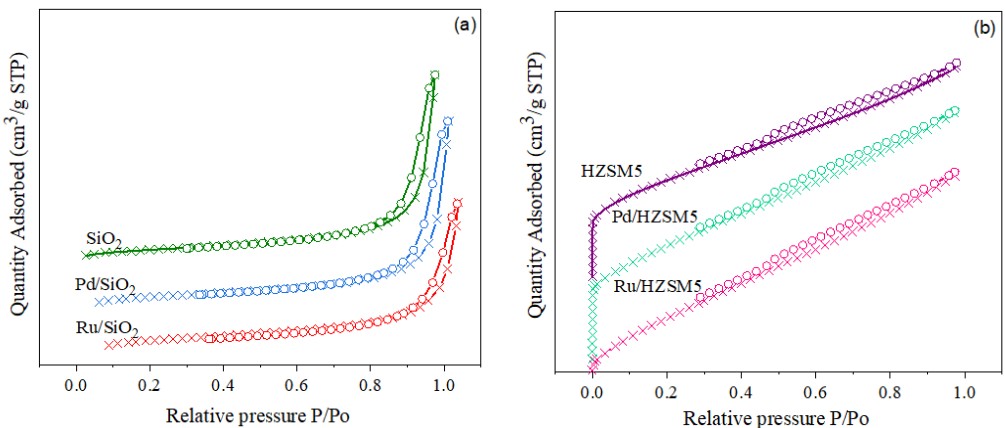

**Figure 1.** $N_2$ adsorption–desorption isotherms of (**a**) $SiO_2$, Pd/$SiO_2$, and Ru/$SiO_2$; (**b**), HZSM5, Pd/HZSM5, and Ru/HZSM5.

The diffractograms of supports and catalysts after reduction under 30 bar of $H_2$ at 673 K for 1 h are presented in Figure 2. The X-ray diffraction patterns of $SiO_2$ showed a broad line at $2\theta = 20°$ corresponding to non-crystalline $SiO_2$, while the diffractogram of HZSM5 exhibited the characteristic diffraction lines of the MFI structure of zeolites (PDF: 00-057-0145). Regarding the reduced catalysts, the diffractograms exhibited the lines observed for the supports. Characteristic lines of metallic Pd at $2\theta = 40.02°$ and $46.49°$ (PDF: 01-087-0638) were observed for Pd/$SiO_2$ and Pd/HZSM5, with Pd crystallite sizes, measured by XRD, of 17 and 22 nm, respectively (Table 1). The characteristic line of metallic Ru at $2\theta = 43.96°$ (PDF: 04-001-1921) could also be observed for both Ru-based catalysts. For Ru/$SiO_2$ and Ru/HZSM5, the Ru crystallite size also evaluated by XRD was 12 and 15 nm, respectively.

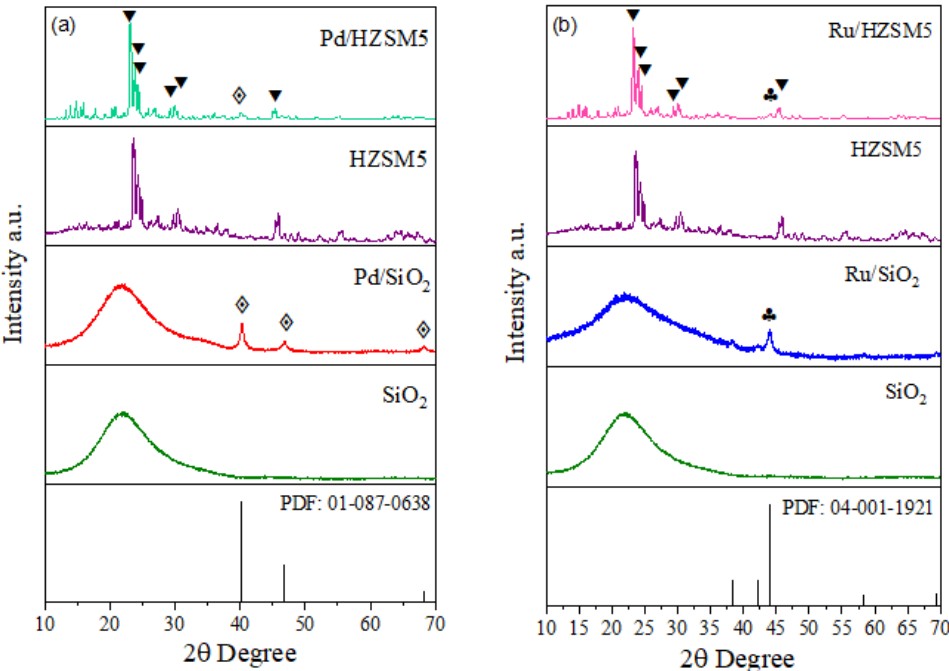

**Figure 2.** Diffractograms of (**a**) supports and Pd-based catalysts, (**b**) supports and Ru-based catalysts. (▼) HZSM5, (◇) $Pd^0$, and (♣) $Ru^0$.

STEM images of $Pd/SiO_2$, $Pd/HZSM5$, $Ru/SiO_2$, and $Ru/HZSM5$, as well as the respective particle size distribution, are shown in Figure 3. Palladium and ruthenium particles had a spherical, or close to spherical, shape. For Pd-supported catalysts, the average Pd particle size varied from 7.2 to 24.5 nm, while for Ru-based materials, the average particle size varied from 8.3 to 12 nm (Table 1). This result is in accordance with the crystallite size estimated by the Scherrer equation, which puts more weight on the particles with the largest crystalline domains. Globally speaking, the size distribution is narrower on silica and narrower for $Ru/HZSM5$. $Pd/HZSM5$ presents the largest particles and the broadest size distribution.

From the STEM results, the metal dispersion was calculated considering ($D = 1.1093/dp \times 100$) for Pd and ($D = 1.3521/dp \times 100$) for Ru. The dispersion of Pd- and Ru-based catalysts varied between 5% and 16% (Table 1).

The $H_2$-TPR of calcined Ru-based catalysts was performed in order to study the reduction of $RuO_2$ on the two supports (Figure 4). One intense peak at 436 K was observed for the TPR profile of $Ru/SiO_2$, while two peaks with Tmax at 393 and 414 K were detected for Ru supported on HZSM5, but in the two cases, they corresponded to the complete reduction of $RuO_2$ to metallic Ru [34–36]. As the reduction of Ru species takes place at low temperatures (<473 K) under diluted $H_2$, the conditions used for catalyst activation (673 K, 30 bar $H_2$) should also lead to a complete reduction of the Ru oxide to metallic Ru.

The $NH_3$-TPD profiles of desorbed ammonia for Pd- and Ru-supported catalysts are shown in Figure 5 As expected, the catalysts supported on $SiO_2$ did not adsorb $NH_3$. Concerning $Pd/HZSM5$ and $Ru/HZSM5$, both catalysts exhibited multiple and poorly resolved peaks between 373 and 773 K, indicating the presence of acidic sites with different strengths. The relative contribution of each individual desorption peak was obtained by decomposition of the TPD profiles, considering the adsorption of three groups of acidic sites of different strength (weak: T < 523 K; medium: 523 < T < 593 K; strong: T > 593 K). The experimental data were fitted using a multiple-Gaussian function. The total density of acidic sites as well as their distribution calculated from $NH_3$-TPD profiles are reported in Table 2. $Pd/HZSM5$ exhibited a higher total density of acid sites (699 μmol/g of $NH_3$) than Ru supported on HZSM5 (521 μmol/g of $NH_3$). However, the distribution is quite similar in both catalysts.

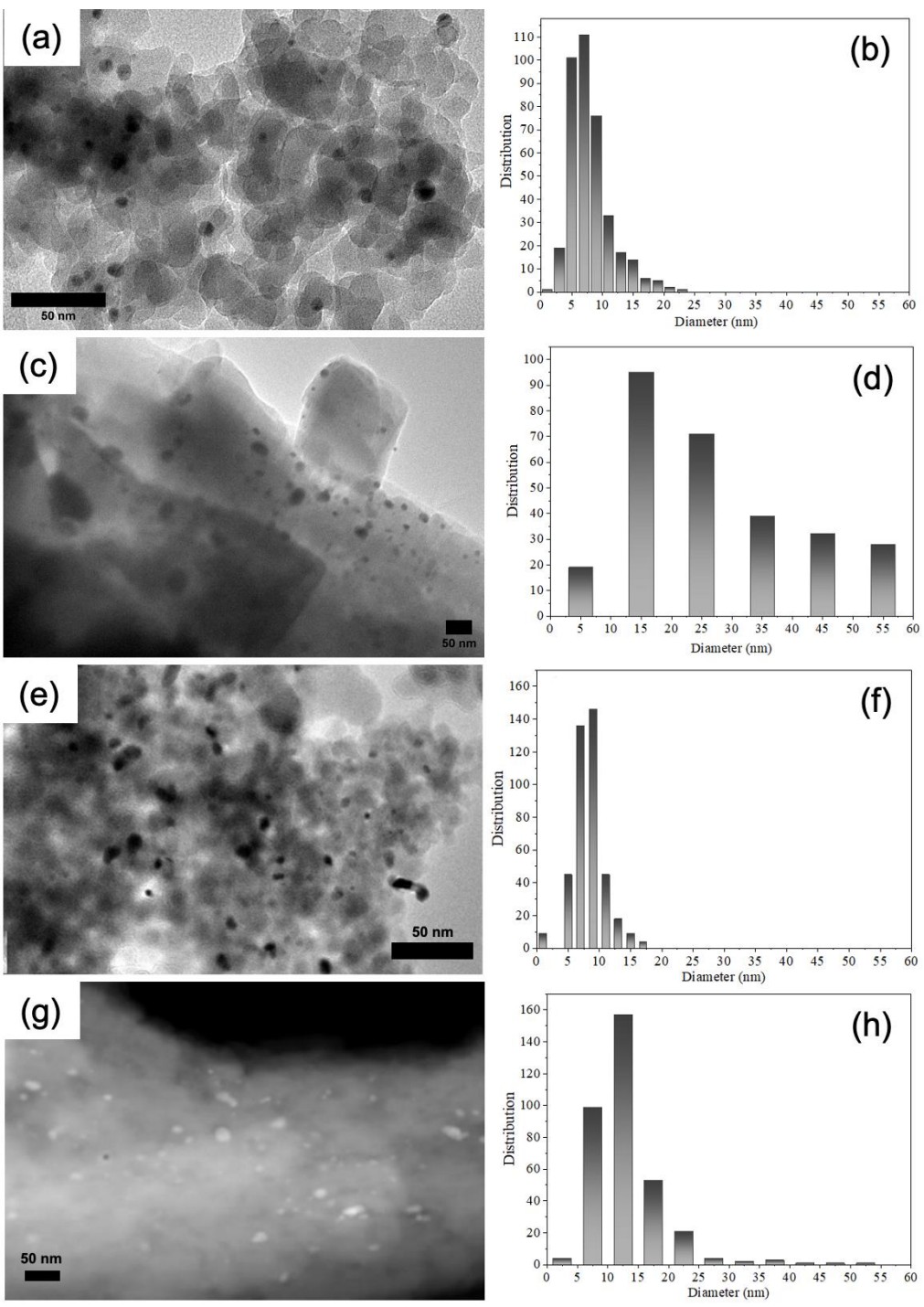

**Figure 3.** TEM images and particle size distribution for (**a,b**) Pd/SiO$_2$, (**c,d**) Pd/HZSM5, (**e,f**) Ru/SiO$_2$, and (**g,h**) Ru/HZSM5. (**a,c,e**) represent bright-field TEM images and (**g**) dark-field TEM images.

**Table 2.** Total amount of ammonia desorbed and acidic sites strength distribution for palladium- and ruthenium-based catalysts.

| Catalysts | Ammonia Desorbed ($\mu$mol g$^{-1}$) | Acidic Sites Strength Distribution (%) | | |
|---|---|---|---|---|
| | | Weak | Medium | Strong |
| Pd/HZSM5 | 699 | 34 | 42 | 24 |
| Ru/HZSM5 | 521 | 31 | 42 | 27 |

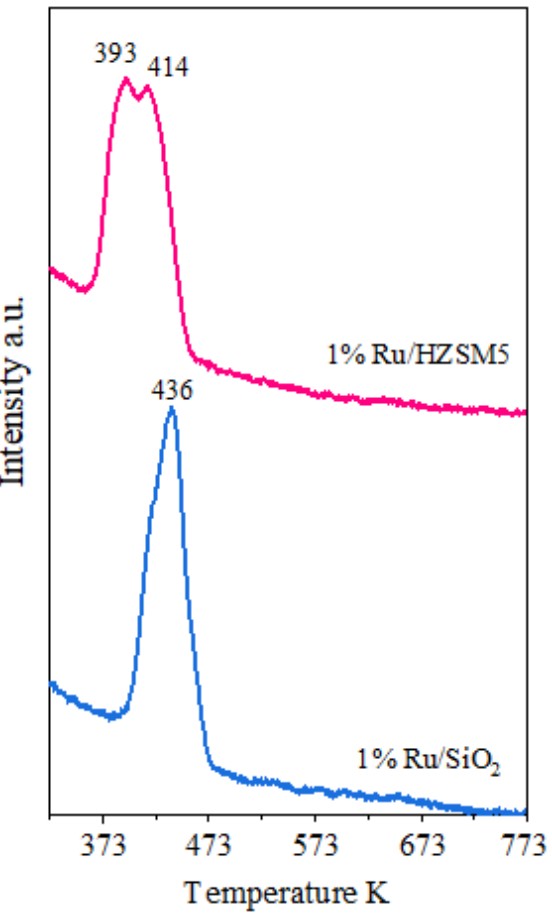

**Figure 4.** TPR profiles of ruthenium-supported catalysts.

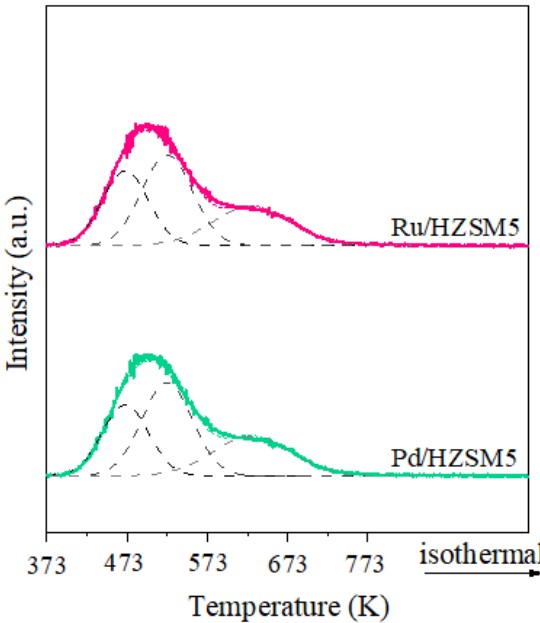

**Figure 5.** NH$_3$-TPD profiles of Pd/HZSM5 and Ru/HZSM5 after reduction (30 bar H$_2$ and 673 K).

### 2.2. Conversion of Phenolic Dimers

In order to evaluate the influence of the type of metal and support, Pd and Ru supported on SiO$_2$ and HZSM5 were used for the conversion of dimeric aryl ethers (benzyl

phenyl ether, phenethoxybenzene, and diphenyl ether) representative of the main ether linkages present in the lignin, α-O-4, β-O-4, and 4-O-5. The reactions were carried out at 503 K, under 18 bar of $H_2$, in decalin for 3 h.

### 2.2.1. Benzyl Phenyl Ether (BPE)

The product distribution for BPE conversion over Pd and Ru supported over $SiO_2$ and HZSM5, as well as over the bare supports, are presented in Table 3. No conversion was detected in the absence of a catalyst.

**Table 3.** Conversion of BPE, carbon balance, and product distribution for Ru- and Pd-based catalysts.

| Entry | Catalysts | $X_{BPE}$ (%) | CB (%) | Yield (%) | | |
|---|---|---|---|---|---|---|
| 1 | $SiO_2$ | 0 | 100 | 0.0 | 0.0 | 0.0 |
| 2 | $Pd/SiO_2$ | 41 | 90 | 18.2 | 13.0 | 0.0 |
| 3 | $Ru/SiO_2$ | 31 | 91 | 6.1 | 5.2 | 10.9 |
| 4 | HZSM5 | 29 | 95 | 0.2 | 4.9 | 19.1 |
| 5 | Pd/HZSM5 | 59 | 91 | 4.7 | 10.9 | 33.6 |
| 6 | Ru/HZSM5 | 62 | 91 | 1.0 | 8.6 | 42.9 |

Reaction conditions: BPE (62 mmol), decalin (15 mL), $SiO_2$ (250 mg), $Pd/SiO_2$ (250 mg), $Ru/SiO_2$ (300 mg), HZSM5 (60 mg), Pd/HZSM5 (60 mg), Ru/HZSM5 (45 mg), 503 K, 18 bar $H_2$, 3 h.

$SiO_2$ did not exhibit activity for BPE conversion, even using a high amount of support (250 mg), due to the absence of acidic sites. When Pd and Ru catalysts supported on $SiO_2$ were used, toluene and phenol were formed through the hydrogenolysis of BPE catalyzed by metal sites, which is in agreement with the literature [10,19,22]. However, in addition to the formation of the hydrogenolysis products, alkylated products, 2- and 4-benzylphenol (2- and 4-BPH), were also detected over $Ru/SiO_2$.

For supported catalysts, the formation of 2- and 4-BPH has only been reported in a few works. He et al. [12] reported their formation over $Ni/SiO_2$ and Ni/HZSM5 at 523 K using undecane as a solvent. However, this was attributed to a solvent effect that could cleave BPE by pyrolysis in the reaction conditions used. In our case, no transformation was detected in the absence of a catalyst. Dou et al. [37] used Co–Zn supported on dealuminated zeolite Hβ for the conversion of BPE under 40 bar of $H_2$ at different temperatures (413–533 K), using methanol as a solvent. At high temperatures (>453 K), a significant yield of 2- and 4-BPH was observed (>40%). According to the authors, they were formed from an alkyl-aryl ether rearrangement catalyzed by Lewis acid sites. Ru/HZSM5 [23] (493 K, 20 bar of $H_2$) and Pd/C in the presence of HZSM5 [29] (473 K, 50 bar of $H_2$) were also used in the literature for the conversion of BPE in *n*-hexane and aqueous phase, respectively. However, these catalysts led to the hydrogenolysis of BPE and to the further hydrogenation and hydrodeoxygenation of the hydrogenolysis products (phenol and toluene). Despite the presence of acidic sites, the formation of 2- and 4-BPH was not reported.

Therefore, the formation of alkylated products has only been reported in the literature either via pyrolysis or by the catalytic action of acidic sites. According to our results of $NH_3$-TPD, no acidity was measured for $Pd/SiO_2$ and $Ru/SiO_2$. Furthermore, the $H_2$-TPR profiles of Ru-based catalysts showed that the Ru oxide was completely reduced at 673 K, excluding the action of Lewis acidic sites associated with unreduced ruthenium oxide. Therefore, these results suggest that the oxophilicity of the metal, i.e., the ability of Ru to bind to the oxygen atom from the substrate, may be involved in the promotion of the alkylation reactions.

The metal oxophilicity is directly associated with the position of the d-band center relative to the Fermi level. As the d-band center is closer to the Fermi level, the anti-bonding orbital of the hybridization between metal and oxygen is further away from the Fermi level

and has low electron occupancy. Therefore, the M-O bond becomes stronger and the metal more oxophilic [38,39]. Duong et al. [39] reported a correlation between the activation energy for the direct deoxygenation (DDO) of m-cresol to toluene and the adsorption energy of oxygen over different metals. They observed that the energy barrier for the DDO of m-cresol increased linearly from Fe to Pt as the metal oxophilicity decreased. For the metals evaluated, the oxophilicity increased in the order Pt < Pd < Rh < Ru < Fe. As a consequence, the DDO was favored over Rh, Ru, and Fe.

The effect of the nature of the metal was previously investigated by our group for the hydrodeoxygenation (HDO) of phenol in the gas phase at 573 K [40]. It was observed that over Pt, Pd, and Rh, phenol was mainly tautomerized and hydrogenated to cyclohexanone/cyclohexanol, whereas the direct dehydroxylation of phenol to benzene was favored for oxophilic metals (Ru, Co, and Ni). Similar results were observed by Tan et al. [41] for the HDO of m-cresol over Pt and Ru supported on $SiO_2$. DFT calculations showed that the direct dehydroxylation of m-cresol to toluene was favored over the oxophilic Ru (001) surface due to the lower energy barrier (98 kJ mol$^{-1}$) compared to Pt (111) (242 kJ mol$^{-1}$). The authors proposed that over Ru sites, the dehydroxylation of m-cresol produces an unsaturated hydrocarbon radical $C_7H_7$*, which can be hydrogenated to form toluene. Over Pt (111), the same intermediate was not formed.

Therefore, we propose that the cleavage of BPE into phenoxy and benzyl radicals occurs not only on acidic sites but also on the surface of Ru due to its high oxophilicity. According to the literature [42], due to the lower hydrogenation ability of Ru compared with Pd, Ru does not easily promote the hydrogenation of phenoxy and benzyl radicals into phenol and toluene, respectively. Thus, the higher concentration of phenoxy and benzyl radicals may favor their recombination into 2- and 4-BPH (Scheme 1).

**Scheme 1.** Reaction route for BPE conversion over Pd- and Ru-based catalysts.

Regarding the results in the presence of acidic sites (Table 3, entries 4–6), 29% of BPE was mainly converted into phenol and, indeed, 2-BPH over bare HZSM5. In addition, toluene appeared as a minor product. For Pd/HZSM5 and Ru/HZSM5, the same products were observed at a similar conversion (59% and 62%, respectively). Compared to HZSM5, the presence of metal particles increased the conversion of BPE by hydrogenolysis to phenol and toluene. In addition, and in agreement with the observations above, a larger formation of 2- and 4-BPH was observed for Ru/HZSM5 compared to Pd/HZSM5.

In the literature, the conversion of BPE using undecane at 523 K under 40 bar of $H_2$ in the absence of a catalyst was indeed reported to produce high amounts of 2- and 4-BPH, which are products of condensation of the toluene moiety to the phenol moiety [12]. The conversion of BPE via free radical reactions at high temperatures (548 K) was also proposed by Kidder et al. [43] for the pyrolysis of BPE confined in mesoporous silica. After cleavage, the phenoxy and benzyl radicals would follow two different reaction pathways: (i) hydrogenation to form phenol and toluene or (ii) recombination to form 2-BPH or 4-BPH. For HZSM5 in undecane, He et al. [12] proposed that the Brønsted acidic sites of HZSM5 donate their proton to the oxygen of the BPE molecule possessing a lone pair of electrons, which weakens the C-O linkage, leading to BPE cleavage into phenoxy and benzyl radicals. When these radicals react with H radicals, phenol and toluene are produced. Finally, phenol can be attacked on the ortho or para position to form 2- and 4-BPH. High selectivity to 2- and 4-BPH was also reported by Yoon et al. [30] for the conversion of BPE over silica-alumina

supports using *n*-decane as solvent. The authors proposed that at low temperatures (523 K), carbocation intermediates were formed on the Brønsted acidic sites of the catalysts, and alkylated products were formed through Claisen rearrangement. They suggested that the formation of alkylated products was promoted by the $H_2$-deficient environment, since silica-alumina does not adsorb and activate hydrogen.

According to our results, the cleavage of BPE over HZSM5 is consistent with these radical reactions proposed by the literature. In the presence of acidic sites, BPE can be broken to form phenoxy and benzyl radicals. Then, these radicals can react with hydrogen to form phenol and toluene, respectively; or benzyl radicals attack phenol to form alkylated products, 2- and 4-BPH (Scheme 1).

### 2.2.2. Phenethoxybenzene (PEB)

The PEB conversion and product yield for Pd- and Ru-based catalysts supported over $SiO_2$ and HZSM5 are shown in Table 4.

**Table 4.** Conversion of PEB, carbon balance, and product distribution for Ru- and Pd-based catalysts.

| Entry | Catalysts | $X_{BPE}$ (%) | CB (%) | Yield (%) | | | | | | | |
|---|---|---|---|---|---|---|---|---|---|---|---|
| 1 | $SiO_2$ | 0 | 99 | 0.0 | 0.0 | 0.0 | 0.0 | 0.0 | 0.0 | 0.0 | 0.0 |
| 2 | $Pd/SiO_2$ | 77 | 87 | 0.0 | 0.0 | 0.0 | 0.0 | 0.0 | 11.0 | 36.0 | 17.1 |
| 3 | $Ru/SiO_2$ | 84 | 61 | 3.4 | 2.6 | 1.5 | 2.4 | 0.0 | 8.1 | 9.0 | 18.3 |
| 4 | HZSM5 | 9 | 97 | 2.6 | 0.0 | 2.4 | 0.0 | 1.0 | 0.0 | 0.0 | 0.0 |
| 5 | Pd/HZSM5 | 79 | 68 | 8.6 | 2.0 | 9.4 | 0.0 | 0.0 | 3.7 | 12.8 | 10.1 |
| 6 | Ru/HZSM5 | 79 | 87 | 1.0 | 0.0 | 8.6 | 0.0 | 56.8 | 0.0 | 0.0 | 0.0 |

Reaction conditions: PEB (31 mmol), decalin (15 mL), $SiO_2$ (250 mg), $Pd/SiO_2$ (250 mg), $Ru/SiO_2$ (250 mg), HZSM5 (44 mg), Pd/HZSM5 (30 mg), Ru/HZSM5 (44 mg), 503 K, 18 bar $H_2$, 3 h.

As observed for BPE, no conversion of PEB was detected in the absence of catalysts nor over $SiO_2$. In the presence of metallic sites, $Pd/SiO_2$ yielded only products from partial and full hydrogenation of PEB aromatic rings: (2-(cyclohexyloxy) ethyl) benzene (2COEB), (2-(cyclohexylethoxy) benzene (2CEB), and (2-(cyclohexylethoxy) cyclohexane (2CEC), and no hydrogenolysis products. Over $Ru/SiO_2$, besides the hydrogenated products formed over $Pd/SiO_2$, ethylbenzene, ethylcyclohexane, phenol, and benzene were also observed as products. These results suggest that over $Ru/SiO_2$, the hydrogenolysis of PEB into phenol and ethylbenzene also occurs to a small extent. Ethylcyclohexane is formed by ethylbenzene hydrogenation, while benzene is produced by hydrogenolysis of the $C_{arom}$-O bond of phenol, both reactions catalyzed by Ru sites.

In the presence of the acidic sites of HZSM5, 9% of PEB was converted into phenol, ethylbenzene, and alkylated products, 2- and 4-phenethyl phenol (2- and 4-PPE), indicating that acidic sites can assist the cleavage of the β-O-4 ether linkage of PEB.

The PEB conversion over bare oxides has indeed been reported in the literature in a non-polar solvent under high hydrogen pressure and temperature. Over Hβ (493 K, 40 bar $H_2$) [14] and USY (618 K and 50 bar $H_2$) [31], zeolites, phenol, and alkylated products (2, 3, and 4-PPE) were the main products observed after PEB cleavage in dodecane. The authors proposed that the cleavage of PEB occurred by H+ addition to produce phenol and 2-phenyleth-1-ylium intermediates, which can attack phenol at the ortho, meta, or para positions to form 2, 3, and 4-PPE, respectively, via a transalkylation reaction catalyzed over the acidic sites. In our work, the product distribution observed for PEB conversion over HZSM5 suggests that the conversion of PEB proceeds by a similar radical route as reported in the literature.

When metallic sites were present on the catalyst, the conversion of PEB increased from 9% (bare HZSM5) to 79%, indicating that metal and acidic sites are required for better performance. Regarding the product distribution, over Pd/HZSM5, 2COEB, 2CEB, 2CEC, phenol, ethylbenzene, and ethylcyclohexane were formed. Alkylated products were not detected, and compared to Pd/SiO$_2$, the hydrogenolysis pathway and the hydrogenation of PEB aromatic rings became more competitive (lesser production of partially or totally hydrogenated dimers). In contrast to Pd/HZSM5, when Ru/HZSM5 was used, the alkylated products from isomerization reactions were by far the main products detected. This can be related to the higher hydrogenation ability of Pd compared to Ru [42].

Salam et al. [31] proposed that the formation of alkylated products by transalkylation reaction occurred simultaneously with the hydrogenolysis of the C$_{aliph}$-O bond of PEB into phenol and ethylbenzene for NiMoS supported over USY zeolite (618 K, 50 bar of H$_2$ in dodecane as solvent). Then, the hydrocracking of alkylated products to produce mono and alkyl phenols (m-cresol and 2- and 4-ethylphenol) occurred. The formation of benzene, toluene, and ethylbenzene was attributed to the HDO of mono and alkyl phenols.

In conclusion, according to our results, the conversion of PEB over Pd and Ru supported on HZSM5 occurs via three parallel reaction routes: (i) the cleavage of PEB into phenol and 2-phenyleth-1-ylium catalyzed by the acidic sites of HZSM5, (ii) the hydrogenolysis of the C$_{aliph}$-O bond of PEB into phenol and ethylbenzene, catalyzed by the metallic sites with the assistance of the acidic sites, and (iii) the partial and full hydrogenation of PEB aromatic rings. The presence of the acidic sites leads to the alkylation reaction between phenol and 2-phenyleth-1-ylium to form 2- and 4-PPE (Scheme 2). Hydrogenation reactions are especially favored by Pd, while Ru contributes to the alkylation pathway.

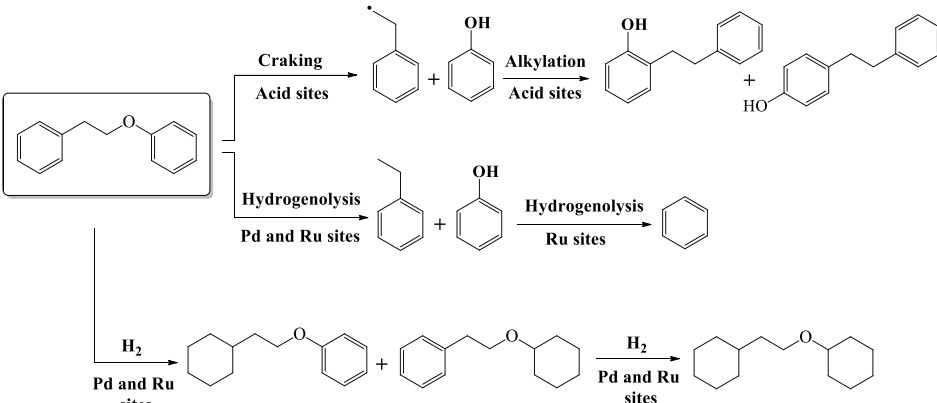

**Scheme 2.** Reaction route for PEB conversion over Pd- and Ru-based catalysts.

2.2.3. Diphenyl Ether (DPE)

The conversion and product distribution for the conversion of DPE over supported Pd and Ru catalysts are shown in Table 5.

As observed for BPE and PEB, SiO$_2$ was not active for the DPE conversion, which was likely due to the absence of acidity. However, in contrast with the former molecules, bare HZSM5 was also unable to convert DPE under the reaction conditions used (503 K, 18 bar H$_2$, decalin). These results are in suitable agreement with the literature, which reported that DPE remained unreacted in the presence of non-acidic [44] and acidic oxides [20,22].

For Pd- and Ru-based catalysts supported on SiO$_2$, no conversion of DPE was observed for Pd/SiO$_2$, whereas 47% of DPE was converted over Ru/SiO$_2$. For Ru/SiO$_2$, cyclohexane (8.9%) was the main product formed, followed by benzene (5.8%), phenol (5.0%), and cyclohexyl phenyl ether (4.2%).

Wu et al. [26] studied the performance of Ru, Pt, and Pd supported on carbon for the HDO of DPE at 393 K in isopropanol as a solvent without an external source of H$_2$. Ru completely converted DPE into cyclohexanol, benzene, and cyclohexane, while Pd/C exhibited the lowest activity, converting less than 5% of DPE into phenol and benzene. The

authors proposed that, unlike Pd, Ru can coordinatively bind to the oxygen atom of DPE, promoting its cleavage into phenoxide and phenyl radicals. Then, the radicals formed over Ru can be further hydrogenated to form benzene and phenol by the active hydrogen from isopropanol on the surface of Ru. This example confirms what we have repeatedly observed here, the affinity of Ru for oxygen-containing substrates and their cleavage into reactive species.

**Table 5.** Conversion of DPE, carbon balance, and product distribution for Ru- and Pd-based catalysts.

| Entry | Catalysts | $X_{BPE}$ (%) | CB (%) | phenol | benzene | cyclohexanol | cyclohexane | methylcyclopentane | cyclohexyl phenyl ether | dicyclohexyl ether |
|---|---|---|---|---|---|---|---|---|---|---|
| | | | | | | | | Yield (%) | | |
| 1 | SiO$_2$ | 0 | 99 | 0.0 | 0.0 | 0.0 | 0.0 | 0.0 | 0.0 | 0.0 |
| 2 | Pd/SiO$_2$ | 0 | 99 | 0.0 | 0.0 | 0.0 | 0.0 | 0.0 | 0.0 | 0.0 |
| 3 | Ru/SiO$_2$ | 47 | 78 | 5.0 | 5.8 | 0.0 | 8.9 | 0.0 | 4.2 | 1.0 |
| 4 | HZSM5 | 0 | 98 | 0.0 | 0.0 | 0.0 | 0.0 | 0.0 | 0.0 | 0.0 |
| 5 | Pd/HZSM5 | 30 | 81 | 0.0 | 0.0 | 1.0 | 5.6 | 4.1 | 0.0 | 0.0 |
| 6 | Ru/HZSM5 | 19 | 100 | 0.0 | 0.0 | 0.0 | 5.5 | 13.8 | 0.0 | 0.0 |

Reaction conditions: Reactant (31 mmol), decalin (15 mL), SiO$_2$ (250 mg), Pd/SiO$_2$ (250 mg), Ru/SiO$_2$ (300 mg), HZSM5 (250 mg), Pd/HZSM5-imp (250 mg), Ru/HZSM5-imp (25 mg), 503 K, 18 bar H$_2$, 3 h.

Regarding the products obtained over Ru/SiO$_2$, DPE can be cleaved into phenoxide and benzyl radicals over the Ru surface but not over Pd sites. Then, the hydrogenation of the radicals formed after DPE cleavage can produce phenol and benzene, respectively. At the same time, the partial hydrogenation of the DPE aromatic rings occurs over Ru/SiO$_2$. Following the DPE cleavage, benzene can also be formed by hydrogenolysis of the C$_{arom}$-O bond of phenol, and in the absence of acidic sites able to assist the dehydration of phenol-derived cyclohexanol, cyclohexane can only be formed via benzene hydrogenation. Therefore, the results suggest that the higher oxophilicity of Ru could favor the hydrogenolysis of the C$_{arom}$-O bond of DPE.

For Pd/HZSM5, 30% of DPE was converted into cyclohexane (5.6%) and methyl cyclopentane (4.1%). A DPE conversion of 19% was achieved for Ru/HZSM5, and although the same products were obtained, the yield of methyl cyclopentane increased from 4.1% to 13.8% over the Ru-based catalyst. In comparison to Ru/SiO$_2$, products from the partial hydrogenation of the DPE aromatic rings were not detected.

Zhao et al. [20] investigated the conversion of DPE over Pd/HZSM5 and Pd–Ni/HZSM5 catalysts (493 K, 20 bar of H$_2$ in *n*-hexane as solvent). For Pd/HZSM5, 70% of DPE was converted into cyclohexane, cyclohexyl phenyl ether, and dicyclohexyl ether, whereas it was completely converted to cyclohexane on the bimetallic catalyst. They proposed that cyclohexyl phenyl ether is an important intermediate in DPE conversion, with two main reaction pathways: (i) hydrogenation to form dicyclohexyl ether or (ii) hydrogenolysis of its C$_{aliph}$-O bond into phenol and cyclohexane. Then, cyclohexanol, formed from phenol hydrogenation or via cyclohexyl phenyl ether cleavage, can be dehydrated to cyclohexene upon catalysis by the acidic sites. Cyclohexane was finally produced by cyclohexene hydrogenation.

According to our results, the product distributions obtained over Pd- and Ru-supported catalysts suggest that DPE was mostly cleaved by hydrogenolysis into phenol and benzene in the presence of metallic and acidic sites. In addition, DPE can also be cleaved into phenoxide and benzyl radicals on the Ru surface due to its oxophilicity. Then, their hydrogenation can produce phenol and benzene, respectively. The formation of cyclohexane can occur via two pathways: (i) through hydrogenation of benzene (non-acidic catalysts) or (ii) via hydrogenation of phenol and subsequent dehydration (acidic catalysts). Finally,

the isomerization of cyclohexane catalyzed by the acidic sites can lead to the formation of methyl cyclopentane [45] (Scheme 3).

**Scheme 3.** Reaction route for DPE conversion over Pd- and Ru-based catalysts.

## 3. Materials and Methods

### 3.1. Catalyst Preparation

SiO$_2$ (fumed silica, average size 0.2–0.3 μm, Sigma Aldrich, Saint Louis, MI, USA, S5505) and HZSM5 (Zeolyst CBV 2314, Conshohocken, PA, USA) were used to prepare Pd- and Ru-supported catalysts. SiO$_2$ was moistened with deionized water and then calcined in a muffle at 673 K (2 K min$^{-1}$) for 3 h. The 2 wt.% Pd (Pd/SiO$_2$ and Pd/HZSM5) and 1 wt.% (Ru/SiO$_2$ and Ru/HZSM5) catalysts were prepared by incipient wetness impregnation of the supports using an aqueous solution of palladium (II) nitrate hydrate (Pd (NO$_3$)$_2$·xH$_2$O, Alfa Aesar, Tewksbury, MA, USA) and ruthenium (III) chloride hydrate (RuCl$_3$·3H$_2$O, Sigma-Aldrich), respectively, with appropriated volume to ensure that all catalysts are loaded with 2% of Pd and 1% of Ru. After impregnation, the samples were dried at 373 K for 12 h and calcined at 673 K (2 K min$^{-1}$) for 3 h in a muffle oven.

### 3.2. Catalysts Characterization

The Pd content was determined by inductively coupled plasma-optic emission spectroscopy (720-ES ICP-OES, Agilent, Santa Clara, CA, USA) with axially viewing and simultaneous CCD detection. The samples (10 mg) were digested in a solution containing 250 μL of HF, 500 μL of H$_2$SO$_4$, and 4 mL of aqua regia (1 HNO$_3$ + 3HCl) and heated to 383 K for 2 h in the autodigestor Vulcan 42 (Questron, Santa Clara, CA, USA). For the Ru-based catalyst, the Ru content was determined by X-ray fluorescence (XRF) using an energy dispersive micro-X-ray Fluorescence spectrometer Bruker (M4 TORNADO, Billerica, MA, USA) with a rhodium X-ray tube, 50 kV/6000 mA (30 W). The specific surface areas of the samples were measured using a Micromeritics ASAP 2020 analyzer (Norcross, GA, USA) by N$_2$ adsorption at 77 K. The X-ray diffraction (XRD, Billerica, MA, USA) patterns were obtained using a BRUKER D8 Advancer diffractometer with Cu Kα radiation ($\lambda$ = 0.1542 nm) over a 2 $\theta$ range of 10–70° at a scan rate of 0.02 °/step and a scan time of 1 s/step. The crystallite size of each sample was estimated using the Scherrer equation, represented by Equation (1), where K is a dimensionless shape factor with a typical value of 0.9 (spherical particles), l is the X-ray wavelength (0.1542 nm), b is the full width at half maximum of the diffraction peak, and q is the Bragg angle corresponding to this diffraction.

$$d = \frac{K\lambda}{\beta cos\theta},$$  (1)

The temperature-programmed reduction under hydrogen ($H_2$-TPR) of Ru-based catalysts was performed in an AutoChem II 2920 (Micromeritics, Norcross, GA, USA) set-up. Before the analysis, the samples (50 mg) were submitted to treatment with $N_2$ at 473 K for 1 h to remove the adsorbed species. Then, the samples were cooled to room temperature and reduced with a mixture containing 5% $H_2$ in Ar (50 mL min$^{-1}$) up to 773 K (5 K min$^{-1}$).

The total density of acidic sites was measured by temperature-programmed desorption of ammonia ($NH_3$-TPD) in AutoChem II equipment (Micromeritics, Norcross, GA, USA), equipped with a thermal conductivity detector and a mass spectrometer. The catalysts (100 mg) were previously reduced under a flow of 40 mL/min in pure $H_2$ with heating of 10 K min$^{-1}$ up to 673 K for 1 h and then purged under He flow (30 mL min$^{-1}$) for 30 min. After reduction, the samples were cooled to 373 K, and the gas was switched to a mixture containing 10% $NH_3$ in He (30 mL min$^{-1}$) for 30 min. The physisorbed ammonia was flushed out with He (50 mL min$^{-1}$) for 2 h. Then, the catalysts were heated under He at 10 K min$^{-1}$ to 773 K. The metal particle size distribution was measured by scanning transmission electron microscopy (STEM, Tecnai, Hillsboro, OR, USA). Before analysis, the samples were reduced as described above and then passivated at room temperature under 5% $O_2/N_2$ (30 mL min$^{-1}$) for 2 h. The metal catalysts were dispersed in isopropanol by ultrasonication, and the suspensions were dropped on holey carbon-coated copper grids. The images were obtained using a MET FEI Titan X-FEG (Tecnai, Hillsboro, OR, USA) with a voltage of 300 kV, equipped with a high-angle annular bright-field and dark-field detector (HAADF), which can provide images with Z contrast with a resolution of 0.7 Å. In addition, these data enabled the calculation of metal dispersion (*D*) by considering the mean diameter of the metal particles (*dp*) based on Equations (2) and (3):

$$dp = \frac{\sum nidi}{\sum ni} \tag{2}$$

$$D = 6 \frac{Vm/am}{dp} \tag{3}$$

where *ni* is the number of spherical particles of diameter *di*, *Vm* is the volume of the metal atom, and *am* is the surface area occupied by an atom. The volume of metal is described in Equation (4):

$$Vm = \frac{M}{\rho\, N_A} \tag{4}$$

where *M* is the atomic mass, $\rho$ the mass density, and $N_A$ is the Avogadro number ($6.022 \times 1023$ mol$^{-1}$). In the case of palladium (*M* = 106.4 g mol$^{-1}$; $\rho$ = 12.02 g cm$^{-3}$) *Vm* = 14.70 Å$^3$ and *am* = 7.93 Å$^2$. For ruthenium (*M* = 101.07 g mol$^{-1}$; $\rho$ = 12.30 g cm$^{-3}$), *Vm* = 13.65 Å$^3$ and *am* = 6.35 Å$^2$. Replacing *Vm* and *am* values on Equation (3), we find the relationship between metal dispersion and the mean diameter particle (nm) for palladium and ruthenium-based catalysts, Equations (5) and (6), respectively:

$$D\,(\%) = \frac{1,1093}{dp} * 100 \tag{5}$$

$$D\,(\%) = \frac{1,3521}{dp} * 100 \tag{6}$$

### 3.3. Catalytic Activity

The reactions were carried out in a 50 mL autoclave Parr reactor (from Parr Instruments, Moline, Illinois 61265, USA) equipped with a magnetic stirrer. Before the catalytic test, all samples were previously reduced under 30 bar $H_2$ (50 ml min$^{-1}$) at 673 K for 1 h in a Screening Pressure Reactor (SPR, UnchainedLab, Pleasanton, CA 94566, USA). In a typical test, the reactor was loaded with the desired quantity of catalysts and 15 mL of reaction mixture containing benzyl phenyl ether (62 mmol L$^{-1}$), phenethoxybenzene

(62 mmol L$^{-1}$), or diphenyl ether (31 mmol L$^{-1}$) in decalin. After the reaction, the liquid samples were filtered and analyzed by GC-FID (Agilent Technologies 5977B MSD, Agilent, Santa Clara, CA, USA), equipped with a CP-Wax 52 CB column (polyethylene glycol phase, 30 m × 250 μm, 0.250 μm). The chromatographic method had a flow rate of 1.7 mL min$^{-1}$ of N$_2$ as the carrier gas, 1:100 split ratio, injector temperature of 523 K, and the temperature program based on five steps: isothermal at 313 K for 3.5 min; ramp up to 423 K at 15 K min$^{-1}$ for 1 min, ramp up to 468 K at 25 K min$^{-1}$ for 2 min, then ramp up 478 K at 2 K min$^{-1}$ for 1 min and ramp up 523 K at 20 K min$^{-1}$ for 10 min. The products were identified by gas chromatography coupled to mass spectrometry GC-MS (Agilent Technologies 5977B MSD, Santa Clara, CA, USA) using the same column and conditions.

### 4. Conclusions

The effect of the nature of the metal (Pd and Ru), as well as the effect of the type of the support (SiO$_2$ and HZSM5), was investigated for the conversion of BPE, PEB, and DPE in the liquid phase. In the presence of SiO$_2$, no conversion was observed for any molecule due to the absence of acidic sites. However, for HZSM5, the selected model molecules were converted to a greater or lesser extent. As the C-O ether bond strength increases, the effect of the support acidity for C-O bond cleavage is less relevant. As a result, the conversion of dimeric aryl ethers over bare HZSM5 followed the order: BPE (29%) >> PEB (9%) > DPE (0%). Furthermore, due to the presence of acidic sites, oxygenated alkylated products were formed over HZSM5.

For the supported catalysts, the product distribution was highly affected by the oxophilicity of Ru. For BPE conversion, alkylated products were produced even in the absence of acidic sites (Ru/SiO$_2$). For PEB, benzene was formed by hydrogenolysis of phenol, catalyzed by the Ru surface over Ru/SiO$_2$. For DPE, while Pd/SiO$_2$ did not exhibit any activity for DPE conversion, 47% of DPE was converted over Ru/SiO$_2$, producing cyclohexane as the main product. Finally, it has been shown that in the case of PEB and DPE, acidic sites could assist the hydrogenolysis pathway and favor the route leading to monomers later hydrogenated into saturated hydrocarbons.

**Author Contributions:** Conceptualization, R.A.R. and F.B.N.; methodology, R.A.R. and R.W.; validation, F.B.N. and E.M.; formal analysis, R.A.R. and F.B.N.; investigation, R.A.R.; resources, R.W.; writing—original draft preparation, R.A.R. and F.B.N.; writing—review and editing, F.B.N., E.M. and R.W.; supervision, F.B.N. and E.M. All authors have read and agreed to the published version of the manuscript.

**Funding:** This study was supported by the French government through the Programme Investissement d'Avenir (I-SITE ULNE/ANR-16-IDEX-0004 ULNE) managed by the Agence Nationale de la Recherche, CNRS, Métropole Européen de Lille (MEL) and Region Hauts-de-France for "CatBioInnov" project are also acknowledged.

**Data Availability Statement:** Not applicable.

**Acknowledgments:** The authors would like to thank Joelle Thuriot for the analysis of ICP and XRF, Ahmed Addad for the images of TEM, and Olivier Gardoll for the experiments of NH$_3$-TPD. Fábio Bellot Noronha also thanks Fundação de Amparo à Pesquisa do Estado do Rio de Janeiro (FAPERJ—E-26/202.783/2017; 200.966/2021), Conselho Nacional de Desenvolvimento Científico e Tecnológico (CNPq—303667/2018-4; 305046/2015-2; 302469/2020-6; 310116/2019-82) for financial support.

**Conflicts of Interest:** The authors declare no conflict of interest.

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
