# Peer review of "Conversion of Dimeric Diaryl Ethers over SiO2- and HZSM5-Supported Pd and Ru Catalysts: A Focus on the Role of the Metal and Acidity"

_catalysts, doi:10.3390/catal13040783_

Round 1
Reviewer 1 Report
Wojcieszak et al. studied the conversion of dimeric diaryl ethers over the combination of different metals and supports to understand the importance of metallic sites and surface acidity. Based on the reaction data and the literature review, they suggested the mechanism for the formation of different products. This represents an interesting transformation, since lignin is a renewable resource, and its use is attractive. The manuscript may be suitable for publication in Catalysts-MDPI after minor revisions.
- Author reported radical mechanisms for the product formation, is that possible to prove experimentally? Usually, the addition of a radical scavenger can suppress the radical intermediate species.
- The TEM pictures showed the largest size of metallic particles on the zeolite material however silica showed better control of small metallic particles. In general, however, the trend is likely to be reversed.
- In conclusion, the conversion of dimeric diaryl ethers followed the order: BPE (29 %) >> PEB (9 %) > DPE (0 %). An explanation should be provided for the observed trend.
- English needs to be polished for better clarity.
Author Response
Wojcieszak et al. studied the conversion of dimeric diaryl ethers over the combination of different metals and supports to understand the importance of metallic sites and surface acidity. Based on the reaction data and the literature review, they suggested the mechanism for the formation of different products. This represents an interesting transformation, since lignin is a renewable resource, and its use is attractive. The manuscript may be suitable for publication in Catalysts-MDPI after minor revisions.
We would like to thank referee 1 for his/her time spent on considering our article for publication. We answered all questions/remarks. Our replays can be found below (in blue)
1. Author reported radical mechanisms for the product formation, is that possible to prove experimentally? Usually, the addition of a radical scavenger can suppress the radical intermediate species.
We would like to thank Referee 1 for this remark. Indeed, the radical hypothesis is already documented in the literature. It could be also be due to the fact that these mechanisms are observed when the support is silica, which excludes a priori the cutting of heterolytic bonds of the R-CH2+-O-R type. Adding a radical inhibitor could be a very interesting, but this is a study in itself. It is in the perspectives of our work and it would be studied in a separate study. In the present article, the important thing is the demonstration of the particular action of Ru, or Pd in conjunction with the acid sites.
2. The TEM pictures showed the largest size of metallic particles on the zeolite material however silica showed better control of small metallic particles. In general, however, the trend is likely to be reversed.
2. All catalysts were prepared using the wet impregnation method. It could be concluded that the metal complexes did not diffuse into the channels and the particles were formed on the outer surface of the support grains. But forming the particles in the channels of the zeolites is not necessarily a relevant strategy, as the molecules we want to transform are large, and the formation of dimers/trimers could contribute to coking the channels and deactivating the catalyst. Zeolite is mainly used here as a carrier for Bronsted acid sites, and not for its structural properties.
3. In conclusion, the conversion of dimeric diaryl ethers followed the order: BPE (29 %) >> PEB (9 %) > DPE (0 %). An explanation should be provided for the observed trend.
The 4-O-5 aryl ether bond (BDE = 314 kJ.mol-1) is stronger than the β-O-4 and α-O-4 aliphatic bonds (BDE = 289 and 218 kJ.mol-1, respectively). In addition, the apparent activation energy for their cleavage follows the order: 97 kJ/mol (4-O-5) > 86 kJ/mol (β-O-4) > 72 kJ/mol (α-O-4) (Fig 3) [53]. Therefore, it is expected that the cleavage of the weaker aliphatic ether bonds (α-O-4 and β-O-4) is favored.
4. English needs to be polished for better clarity.
The English was checked.
Reviewer 2 Report
This manuscript was too nomarl to use such kinds catalysts. Furthermore the authors should focus more on the conversion of raw or real lignin. Therefore, I donnot suggest the publication of this manuscript.
Author Response
This manuscript was too normal to use such kinds catalysts. Furthermore, the authors should focus more on the conversion of raw or real lignin. Therefore, I do not suggest the publication of this manuscript.
We would like to thank Referee 2 for his/her time spent on considering our article for publication. We fully agree that the conversion of raw/real lignin is a really hot topic. In addition, the most difficult in the conversion of a real lignin is the characterization of the formed products and understanding of the reaction pathways. This is why we started with the model lignin compounds. We tried to work with the catalysts that could be easily prepared and using industrially acceptable preparation method.
Reviewer 3 Report
Comments to the Authors
In this work, Rafael and co-authors investigated the effect of the nature of the metal (Pd and Ru), as well as the effect of the type of the supports (SiO2 and HZSM5) for the conversion of benzyl phenyl ether (BPE), phenethoxybenzene (PEB) and diphenyl ether (DPE) in the liquid phase. The experiments have strongly evidenced the conclusions, and I think this work is interesting. In addition, this manuscript have been well organized, thus, I recommend its publication after addressing the following issues.
1. Standard PDF card should be provied in the XRD patterns (Figure 2).
2. High-resolution transmission electron microscopy (HRTEM) image is necessary for analyzing the structure of catalysts.
3. It is best to provide some perspectives in the conclusion section.
4. A highly related review shoud be cited (Small Sci. 2021, 1, 2100061), and some advanced references about Pd/Ru-based catalysts are recommended, such as J. Mater. Chem. A 2019, 7, 20151-20157.
Author Response
In this work, Rafael and co-authors investigated the effect of the nature of the metal (Pd and Ru), as well as the effect of the type of the supports (SiO2 and HZSM5) for the conversion of benzyl phenyl ether (BPE), phenethoxybenzene (PEB) and diphenyl ether (DPE) in the liquid phase. The experiments have strongly evidenced the conclusions, and I think this work is interesting. In addition, this manuscript have been well organized, thus, I recommend its publication after addressing the following issues.
We would like to thank Referee 3 for his/her time spent on considering our article for publication. We answered all questions/remarks. Our replays can be found below (in blue)
- Standard PDF card should be provided in the XRD patterns (Figure 2).
PDF card was added to the Figure 2
- High-resolution transmission electron microscopy (HRTEM) image is necessary for analyzing the structure of catalysts.
Unfortunately, we do not have high resolution TEM images. However, even with the present resolution we can clearly see differences in the metal particles sizes for Pd and Ru. This in our opinion is the most important conclusion from the TEM analysis.
- It is best to provide some perspectives in the conclusion section.
In order to maximize the yield of deoxygenated products and to disfavor side reactions, further research is needed on the design of catalysts as well as catalytic system to turn the hydrodeoxygenation reactions more efficient. Acidic supports were required to produced deoxygenated products but from the existing literature it was not possible to generalize the effect of the type of metal. Therefore, it is essential to develop studies to deeper understand the role of acidic and oxophilic sites in HDO reactions. In addition, tests with mixture of different ethers could also give some insight in the mechanism and possible competitive adsorption of substrates. The final point will be tests of the real lignin.
- A highly related review shoud be cited (Small Sci.2021, 1, 2100061), and some advanced references about Pd/Ru-based catalysts are recommended, such as J. Mater. Chem. A2019, 7, 20151-20157.
We would like to thank Referee 3 for his/her valuable remarks. The cited articles are indeed highly related to our work. They have been added to the introduction part.
Round 2
Reviewer 2 Report
The revised version could be acceptable in current form.
Author Response
We would like thank the referee 2 for time spent to consider our manuscript for publication.